# Neurotransmission Sex Dichotomy in the Rat Hypothalamic Paraventricular Nucleus in Healthy and Infantile Spasm Model

**DOI:** 10.3390/cimb47050380

**Published:** 2025-05-21

**Authors:** Dumitru Andrei Iacobas, Jana Veliskova, Tamar Chachua, Chian-Ru Chern, Kayla Vieira, Sanda Iacobas, Libor Velíšek

**Affiliations:** 1Personalized Genomics Laboratory, Undergraduate Medical Academy, School of Public and Allied Health, Prairie View A&M University, Prairie View, TX 77446, USA; 2Departments of Cell Biology & Anatomy, Obstetrics and Gynecology, and Neurology, New York Medical College, Valhalla, NY 10595, USA; jana_veliskova@nymc.edu; 3Department of Cell Biology & Anatomy, New York Medical College, Valhalla, NY 10595, USA; tamarchachua@gmail.com (T.C.); chianru_chern@nymc.edu (C.-R.C.); kvieira2@student.nymc.edu (K.V.); 4Department of Pathology, New York Medical College, Valhalla, NY 10595, USA; sandaiacobas@gmail.com; 5Departments of Cell Biology & Anatomy, Pediatrics, and Neurology, New York Medical College, Valhalla, NY 10595, USA; libor_velisek@nymc.edu

**Keywords:** Adcy5, cholinergic synapse, dopaminergic synapse, GABAergic synapse, glutamatergic synapse, serotonergic synapse, synaptic vesicle cycle, transcriptomic network

## Abstract

We profiled the gene expressions in the hypothalamic paraventricular nuclei of 12 male and 12 female pups from a standard rat model of infantile spasms to determine the sex dichotomy of the neurotransmission genomic fabrics. Infantile spasms were triggered in rat pups prenatally primed with two doses of betamethasone followed by the postnatal repeated administration of N-methyl-D-aspartic acid to induce spasms. Publicly available microarray data were used to characterize each gene in each condition for both sexes by the independent transcriptomic features: average expression level, control of the transcript abundance, and expression correlation with every other gene. This study revealed substantial sex differences in the expression level, control, and inter-coordination of the investigated genes among the studied groups. The transcriptomic differences assist in providing a molecular explanation of the behavioral differences and development of infantile epilepsy spasm syndrome in the two sexes.

## 1. Introduction

This report presents an analysis of experimental data on the hypothalamic paraventricular nucleus (PVN) and complements a large study aiming to determine the consequences of infantile spasms (infantile epilepsy spasm syndrome; IESS) on the neurotransmission transcriptome within the hypothalamic nuclei in a male and female rat model of IESS. The analyses encompassed the KEGG-constructed pathways: the synaptic vesicle cycle (SVC) [1] and glutamatergic (GLU) [2], GABAergic (GABA) [3], cholinergic (ACh) [4], dopaminergic (DA) [5], and serotonergic (5HT) [6] synapses. Previously, we have shown that prenatal priming with betamethasone increases the occurrence of clinical spasms in the prenatal betamethasone–postnatal N-methyl-D-aspartic acid (NMDA) model and identified the activation of several hypothalamic nuclei (i.e., arcuate nucleus and paraventricular nucleus) during the spasms [7,8]. Thus, we first studied the spasm-induced transcriptomic alterations in the hypothalamic arcuate nucleus (ARC) and the efficacy of two anti-IESS treatments [9,10].

IESS (formerly infantile spasms, also West Syndrome) represents a unique and devastating seizure syndrome of infancy [11,12]. IESS affects one out of each 3200–3400 infants with a yearly occurrence of approximately 1700 new cases in the US [13]. The syndrome consists of (1) characteristic epileptic spasms during infancy (3–24 months of age), (2) interictal hypsarrhythmia (large amplitude and asynchronous waves) on the electroencephalogram (EEG), and (3) developmental or psychomotor arrest/delay [13]. IESS-specific FDA-approved first-line therapy is hormonal (adrenocorticotropin; ACTH, or corticosteroids such as prednisone, prednisolone, or methylprednisolone) or vigabatrin [14,15,16]. However, these treatments neither fully alleviate the condition nor are they free of serious side effects [17,18]. Many patients with IESS may die within the first years of life or suffer from permanent developmental deficits [19]. This poor prognosis necessitates a search for novel treatment targets. Interestingly, IESS has sex preponderance affecting more boys than girls (from 1.5:1.0 to 1.1:1.0) [20,21,22].

We have developed a model of IESS in infant rats, which includes prenatal priming with betamethasone and an early postnatal trigger of spasms by NMDA [7], which can be repetitive [23]. The spasms triggered by NMDA in the betamethasone-primed rat brain are tightly linked to early development, are semiologically similar to spasms observed in human patients with IESS, share similar EEG patterns, and respond to ACTH, corticosteroids (methylprednisolone), or vigabatrin treatment [23,24,25]. This model has been independently reproduced and validated [26,27,28]. The disorganized EEG recordings seen in this model, as well as in the human condition of IESS [12], indicate altered brain circuitry, presumably caused by impaired inter-neuronal communication via neurotransmitters [9]. Our previous imaging studies indicated that the hypothalamus represents a critical hub that may participate in the control of spasms [7].

Among the brain regions, the hypothalamus is responsible for performing numerous neuroendocrine functions through the neuronal networks located in its specialized nuclei [15]. The paired paraventricular nuclei of the hypothalamus, located in the anterior hypothalamus adjacent to the sides of the third ventricle, are central to regulating the stress response and emotions leading to addictive behavior [14,15,16], as well as to promoting satiety [17,18] through excitatory synaptic transmission circuits [17,18]. The PVN is also critical for the synthesis of vasopressin to regulate blood pressure and oxytocin for the control of stress responses and has connections to the brain stem to control sympathetic circuitry [19,29].

The PVN consists of an intricate network of neurons interconnected by cholinergic [20], dopaminergic [21], GABAergic [22], glutamatergic [7], and serotonergic [23] synapses, each classified according to the neurotransmitter used in the synaptic vesicle cycle [24]. GLU is a primary excitatory neurotransmitter [26] and GABA is the most abundant inhibitory [27] synapse in the mammalian central nervous system, and thus, both are associated with the fundamental functions of the nervous system [28,29]. Other neurotransmitters and their corresponding synapses also play important roles in behavior. Accordingly, the ACh synapse facilitates learning, memory, and attention [30,31], the DA synapse controls learning, memory, motivation, and reward [32,33,34], and the 5HT synapse is involved in learning and memory, emotion, abnormal mood, and cognition [35,36,37].

Given the inextricable link between synaptic sex-specific brain organization and behavior, the development of neurological diseases suggests potentially distinct brain neuronal wiring, most likely related to the organizational effects of sex hormones [38,39,40]. Among others, we have previously reported a substantial sex dichotomy in the gene networking and topology of the rat hypothalamic cytoskeleton [41], as well as in the myelination [41] and GLU and GABA synapse genomic fabrics. We defined the genomic fabric of a functional pathway as the most inter-coordinated and stably expressed gene network whose encoded proteins are responsible for that function [8].

In the present study, at the time of the animal sacrifice on postnatal day (P) 16, their developmentally programmed sex differences are already irreversible. Although the gonads become active in males and females around P28-P30 (rat puberty), the males are exposed to gonadal steroid surges prenatally, and most importantly, to the neonatal testosterone surge between P0 and P5. The role of sex hormones in modulating brain activity in epilepsy is well documented (e.g., [30,31,42,43])

## 2. Materials and Methods

### 2.1. Animals, Treatments, and Tissue Collection

We used the offspring of timed-pregnant Sprague Dawley rats (Taconic Farms, Germantown, NY, USA), purchased on gestational day 8 (G8). Dams were housed in our AAALAC-accredited animal facility with free access to chow and water on a 12 h light–dark cycle (lights on at 7:00 a.m.). All experiments were approved by the New York Medical College Institutional Animal Care and Use Committee (IACUC) and conform to the NIH Guide for the Care and Use of Laboratory Animals, 8th edition.

On their G15, 10 pregnant females were injected twice with either saline or 0.4 mg/kg of betamethasone phosphate (Sigma-Aldrich, St. Louis, MO, USA). After birth on G23 (designated as P0), pups were weighed and sexed. Some of the prenatal betamethasone-primed male and female pups received N-methyl-D-aspartic acid (NMDA) on P12, P13, and P15 to trigger spasms. Spasms were followed for 60 min after the trigger. The remainder of the betamethasone-primed male and female pups were injected with the corresponding volume of saline for control (no spasms). Animals were euthanized on P16 under deep CO_2_ inhalation anesthesia, the pups were quickly perfused with ice-cold saline, the brains were removed, and the PVNs were dissected. The tissue was immediately snap frozen in dry ice for further processing. There were 6 groups labeled by three letters where the first is the sex (M/F), the second is the priming saline/betamethasone (S/B), and the third is the presence (Y/N) of spasms: MSN, FSN, MBN, FBN, MBY, and FBY (or analogously SN, BN, and BY for each sex). No more than two male and two female pups from each litter entered the experiments so that each group was composed of pups collected from two mothers.

### 2.2. Microarray and Data Processing

Profiling the gene expressions of technical replicates convinced us that Agilent microarrays (Agilent, Santa Clara, CA, USA) were not only better priced but, with our improved wet protocol, had a lower technical noise than the Illumina MiSeq and NextSeq 550 (Illumina, San Diego, CA, USA) available to us at the time. Most genomists use 3 biological replicas per condition, and we used 4 (×3 conditions = 12 per sex), which considerably increased the statistical relevance. However, more than 4 replicas would only increase the experiment costs without improving the statistical significance because of the inherent technical noise of the method.

Total RNA was extracted with the Qiagen RNeasy mini-kit (Qiagen, Venlo, Netherlands), the concentration was determined with a NanoDrop ND-2000 Spectrophotometer (ThermoFisher Scientific, Thessalonikis, Greece), and purity was determined with the Agilent RNA 6000 Nano kit in an Agilent 2100 Bioanalyzer. Total RNA was reverse transcribed in the presence of Cy3/Cy5 dUTP, and the incorporation of the fluorescent tags was determined with the NanoDrop. Fluorescently labeled RNAs were hybridized with the dual-mode Agilent-028282 Whole Rat Genome Microarray 4x44K v3, printed with 4 arrays of 45,220 spots with a total of 37,520 sequences, out of which 30,100 were mapped to at least one gene (16,066 unigenes). The arrays were scanned and primarily analyzed with an Agilent G4900DA SureScan Microarray Scanner Bundle (dual laser scanner + PC data system + Feature Extraction Software G4464AA). The wet protocol and the raw data were deposited in the publicly accessible Gene Expression Omnibus databases [44,45,46]. All spots affected by local corruption or with foreground fluorescence less than twice the background in one microarray were disregarded, and background-subtracted foreground fluorescence signals were normalized to the median, and the results were averaged for every set of spots redundantly probing the same gene. Normalization to the median gene expression provides a comparable expression of individual genes across biological replicas, otherwise affected by the errors in sizing the samples.

According to our Genomic Fabric Paradigm [47], the expression of each quantifiable gene is characterizable by three independent measures: *AVE* = average expression level (1), *REC* = Relative Expression Control (2), and *COR* = expression correlation with each other gene (3). *AVE* shows how much the gene is expressed with respect to all other genes, *REC* indicates the influence of the homeostatic mechanisms, while *COR* justifies the gene networking in functional pathways. Together, *AVE*, *REC,* and *COR*, defined by the below formulas, provide the most theoretically possible comprehensive characterization of the transcriptome topology:∀i,j=1 ÷N & ∀s=MSN, FSN, MBN, FBN, MBY,FBY(1)$${AVE}_{i}^{\left(s\right)}\equiv \;\frac{1}{4}\sum_{k=1}^{4}\left({\frac{\propto_{i;k}^{\left(c\right)}}{\langle \alpha_{i;k}^{\left(c\right)}\rangle }}_{i;k}^{\left(c\right)}\right)\;with\;a_{i}^{\left(c\right)}=\frac{1}{4}\sum_{k=1}^{4}\propto_{i;k}^{\left(c\right)}$$

$\propto_{i;k}^{\left(c\right)}$ is the background-subtracted fluorescence of the spot probing for which the gene in replica *k* (=1, 2, 3, 4) of sample *s*, $\langle \alpha_{i;k}^{\left(c\right)}\rangle$ is the net fluorescence of the median gene, and $a_{i}^{\left(c\right)}$ is the average net fluorescence over all biological replicas.(2)$${REC}_{i}^{\left(c\right)}\equiv {log}_{2}\left(\frac{\langle {REV}_{i}^{\left(c\right)}\rangle }{{REV}_{i}^{\left(c\right)}}\right),$$
where(2a)$${RCS}_{i}^{\left(c\right)}\equiv \frac{\langle {REV}_{i}^{\left(c\right)}\rangle }{{REV}_{i}^{\left(c\right)}}=\;the\;relative\;control\;strength$$(2b)$$\langle {REV}_{i}^{\left(c\right)}\rangle \;is\;the\;median\;REV$$(2c)$${REV}_{i}^{\left(c\right)}\equiv \;\frac{\sigma_{i}^{\left(c\right)}}{2{AVE}_{i}^{\left(c\right)}}\left(\sqrt{\frac{r_{i}}{\chi^{2}\left(\beta ;r_{i}\right)}}-\sqrt{\frac{r_{i}}{\chi^{2}\left(1-\beta ;r_{i}\right)}}\right)$$

*REV* is computed using the midinterval of the χ^2^ distribution of the coefficient of variation in the normalized expression levels across biological replicas. *REV* shows how spread the expression levels are across the biological replicas of that particular type of sample. The expression controls in two conditions of the same sex or one condition in both sexes were compared through the fold-change (*FC*), negative for down-regulation:(2d)$${FC}_{i}^{\left(compared\;vs\;reference\right)}=\left\{\begin{array}{c} \frac{{RCS}_{i}^{\left(compared\right)}}{{RCS}_{i}^{\left(reference\right)}}\;,\;if\;{RCS}_{i}^{\left(compared\right)}>{RCS}_{i}^{\left(reference\right)}\;\\ -\frac{{RCS}_{i}^{\left(reference\right)}}{{RCS}_{i}^{\left(compared\right)}},\;if\;{RCS}_{i}^{\left(compared\right)}\leq {RCS}_{i}^{\left(reference\right)} \end{array}\right.$$

*COR* is the pair-wise Pearson correlation coefficient of the (log_2_) of the normalized expression levels of the two genes [48].(3)$${COR}_{i,j}^{\left(c\right)}\equiv \;correl\;\left(\log_{2}\left({\frac{\propto_{i;k}^{\left(c\right)}}{\langle \alpha_{i;k}^{\left(c\right)}\rangle }}_{i;k}^{\left(c\right)}\right),\;\log_{2}\left({\frac{\propto_{j;k}^{\left(c\right)}}{\langle \alpha_{j;k}^{\left(c\right)}\rangle }}_{i;k}^{\left(c\right)}\right)\right)$$

*COR* analysis [48] was used to identify the *p* < 0.05 significant inter-gene synergistic/antagonistic/independent expressions. This analysis is the prerequisite to determine the most active transcriptomic networks interlinking (here) the genes involved in neurotransmission. It is important to remember that the statistically significant positive correlation means that the expression of either gene stimulates the expression of the other, the negative correlation points out the opposite tendency, while independence reveals the total decoupling of the encoded products of the two genes. *COR* analysis can also be used to test the validity of the wiring in the KEGG-constructed pathways under normal conditions (M/F)SN and quantify their remodeling in the imposed conditions (M/F)BN and (M/F)BY. Moreover, by comparing the results in males and females, one can find whether the documented effects of the sex hormones on neurotransmission [49,50,51] should include the differences in the expression control and organization of gene transcriptomic networks beyond differential expression levels.

The transcriptomic influential powers of individual genes computed using the measure termed “Gene Commanding Height”, *GCH*, that combines the expression control and the median of the expression coordination with all other genes:(4)$${GCH}_{i}^{\left(c\right)}\equiv exp\left({REC}_{i}^{\left(c\right)}+{\langle {\left({2COR}_{i,j}^{\left(c\right)}\right)}^{2}\rangle }_{j}\right)$$

The top gene (highest GCH) in each condition was termed the Gene Master Regulator (GMR). Pending of the type of genetic manipulation, the GMR might be the most legitimate target for the gene therapy aiming to destroy or to stimulate the proliferation of the most abundant clone in that condition [52].

The expression of a gene is considered significantly regulated when comparing BN with SN and BY with BN for each sex or significantly different when comparing the two sexes within the same condition if it satisfies the composite criterion (4). For the absolute fold-change |*x*| and the *p*-value of the heteroscedastic *t*-test of means’ equality [53],(5)$$\left\{\begin{array}{c} \left|x_{i}^{\left(compared\;vs\;reference\right)}\right|>{CUT}_{i}^{\left(compared\;vs\;reference\right)}\\ p_{i}^{\left(compared\;vs\;reference\right)}<0.05 \end{array}\right.$$
where(5a)$$x_{i}^{\left(compared\;vs\;reference\right)}=\left\{\begin{array}{c} \frac{{AVE}_{i}^{\left(compared\right)}}{{AVE}_{i}^{\left(reference\right)}}\;,\;if\;{AVE}_{i}^{\left(compared\right)}>{AVE}_{i}^{\left(reference\right)}\;\\ -\frac{{AVE}_{i}^{\left(reference\right)}}{{AVE}_{i}^{\left(compared\right)}},\;if\;{AVE}_{i}^{\left(compared\right)}\leq {AVE}_{i}^{\left(reference\right)} \end{array}\right.$$(5b)$${CUT}_{i}^{\left(compared\;vs\;reference\right)}=1+\sqrt{\frac{2}{100}\left({\left({REV}_{i}^{\left(compared\right)}\right)}^{2}+{\left({REV}_{i}^{\left(reference\right)}\right)}^{2}\right)}$$

### 2.3. KEGG-Constructed Functional Neurotransmission Pathways

The analyses were directed toward genes associated with the KEGG (Kyoto Encyclopedia of Genes and Genomes, [54])-constructed pathways: synaptic vesicle cycle (SVC, denoted by “0” in the column “Path” of tables below), and glutamatergic (GLU, “1”), GABAergic (GABA, “2”), cholinergic (ACh, “3”), dopaminergic (DA, “4”), and serotonergic (5HT, “5”) synapse [1,2,3,4,5,6]. We were able to quantify properly in all samples: 69 out of 80 of the KEGG-listed SVC genes, 90 out of 115 GLU genes, 67 out of 90 GABA genes, 80 out of 113 ACh genes, 111 out of 132 DA genes, and 71 out of 130 5HT genes. The pathways are partially overlapping, with several genes shared by two or more pathways. For instance, *Gnai2* is part of all five types of synapses and *Mapk3* of the glutamatergic, cholinergic, and serotonergic synapses.

## 3. Results

### 3.1. There Is Little Sex Dichotomy of the Most Highly Expressed Neurotransmission Genes

Table 1 presents the top five genes associated with the functional pathways, SVC (“0”), GLU (“1”), GABA (“2”), ACh (“3”), DA (“4”), and 5HT (“5”), which exhibited the largest expression levels in each sex in all three conditions. Of note is the almost unchanged gene hierarchy according to their normalized (to the median gene) expression levels, although their expression was a little higher in females (see Appendix A for the ratio “×” of the (M/F) expression levels according to definition (5a)). For instance, with respect to the median gene expression, the DA gene *Caly* has 126× more transcripts than the median gene in male SN (but 195× in female SN), 139× in male BN (but 143× in female BN), and 126× in male BY (but 155× in female BY). For comparison, the table also includes the non-neurotransmission genes (no number in the “Path” column) with the largest expression levels (*Cst3* and *Rpl41*), indicating that the neurotransmission genes are among the most highly expressed in the PVN transcriptome. However, although these top five genes in the analyzed neurotransmission functional pathways were practically not differentially expressed, other neurotransmission genes presented statistically significant differential expression between the two sexes. However, it is interesting to note the differences among the three conditions for each sex.

### 3.2. Sex Dichotomy in the Expression Control and Alteration by IESS

Table 2 and Table 3 present the most and the least controlled neurotransmission genes in the paraventricular nodes of both sexes in all three conditions as quantified by the Relative Expression Control (*REC*) score. Positive *REC*s indicate how many times a gene is under stricter control than the median gene, while a negative *REC* indicates how many times the gene is less controlled than the median of that group of samples.

The most controlled genes (highest positive *REC*s) are most likely critical for cell survival and phenotypic expression, while the least controlled genes might be essential for the cell adaptation to the slight environmental changes like those differentiating the biological replicas. Appendix A list the ratios of the relative control strength (*RCS*) scores of the same genes between the two sexes in all three conditions.

One of the most controlled neurotransmission genes in the IESS female PVN, *Gng10*, is part of all five types of synapses analyzed in this study. Of note are the substantial differences between the two sexes in all three conditions, as well as the switch from very controlled in (M/F)SN to less controlled in (M/F)BN of *Atp6v0b*, *Calml4*, and *Atp6v1b*. Interestingly, the dopaminergic gene, *Th*, with a very loose control in male SN, became strictly controlled in the condition of spasms.

We believe that the reason why the expressions of some genes were left free to fluctuate is to provide adaptation to the changes in the environmental conditions. Thus, *Gnb3* and *Gng5* were the most flexibly expressed genes in the BN condition of both male and female PVNs five synapse pathways.

Figure 1 illustrates the substantial sex differences in the gene expression control by plotting the *REC*s of the most (1a) and the least (1b) controlled genes in the three conditions of the hypothalamic paraventricular nuclei of rat pups subjected to the three conditions. It is remarkable that genes such as *Pik3r5*, *ThTrpc1*, *Erg28*, *Mapk10*, and *Creb3* are over-controlled in some conditions and less controlled in others, indicating differences in the regulatory homeostatic mechanisms.

### 3.3. Sex Differences in the Unaltered State of the Six Neurotransmission Pathways

Figure 2 presents the statistically significant (i.e., satisfying the composite criterion of *p* < 0.05 + absolute expression ratio > CUT) differential expression of the SVC (a), GLU (b), GABA (c), ACh (d), DA (e), and 5HT (f) pathways’ genes between males and females in the unaltered state (SN) of the paraventricular hypothalamic nucleus. In this figure, the female transcriptome is the reference and the male is the referred, so that the red background indicates a significantly higher expression in males while a green background specifies higher expression in females.

### 3.4. Sex Differences Between the Significantly Regulated SVC Genes in the PVNs by the Induction of Spasms in the Betamethasone-Primed Rats

Figure 3 presents the statistically significant (*p* < 0.05, absolute fold-change > CUT) regulations in the SVC functional pathway following the induction of the spasms in the betamethasone-primed state (BYS/BNS) of the PVNs of male and female rats.

Out of the sixty-nine quantified SVC genes, one was up-regulated in males while four were up-regulated in females. Three SVC genes were down-regulated in males versus two in females. Interestingly, two down-regulated genes in males, *Atp2a2* and *Cplx2,* were up-regulated in females, indicating opposite effects of triggering the infantile spasms in the two sexes.

### 3.5. Sex Differences Between the Significantly Regulated GLU Genes in the PVN by the Induction of Infantile Spasms in the Betamethasone-Primed Rats

Figure 4 presents the statistically significant (*p* < 0.05, absolute fold-change > CUT) regulations in the GLU functional pathway following the induction of the spasms in the betamethasone-primed state (BYS/BNS) of the PVNs of male and female rats. Out of the ninety quantified GLU genes, four genes were up-regulated and three were down-regulated in males, compared to two up-regulated and one down-regulated in females. Interestingly, *Adcy5*, included in all five synaptic pathways, was down-regulated by IS in males but up-regulated in females, indicating opposite effects of IS on the two sexes.

### 3.6. Sex Differences Between the Significantly Regulated GABA Genes in the PVN by the Induction of Infantile Spasms in the Betamethasone-Primed Rats

Figure 5 presents the statistically significant (*p* < 0.05, absolute fold-change > CUT) regulations in the GABA functional pathway following the induction of the infantile spasms in the betamethasone-primed state (BYS/BNS) of the PVNs of male and female rats. Out of the sixty-nine quantified GABA genes, four were up-regulated and two were down-regulated in males, compared to one up-regulated and four down-regulated in females, with *Adcy5* being down-regulated in males but up-regulated in females.

### 3.7. Sex Dichotomy of the Genes’ Transcriptomic Networks

Figure 6 shows the significant differences between the two sexes in the correlated expressions of SVC genes. Panel (a) presents the SVC gene pairs that are oppositely correlated in the two sexes, while, as shown in panels (b) and (c), several independently expressed gene pairs in one sex became significantly synergistically or antagonistically expressed in the other.

Figure 7 shows the significant differences between the two sexes in the correlated expressions of GLU genes. Panel (a) presents the genes that are oppositely correlated in the two sexes. Thus, four antagonistically expressed gene pairs in males were switched to synergistically expressed in females (*Gls2*—*Gng13*, *Gnao1*—*Gria2*, *Gng5*—*Itpr1*, *Grm2*—*Slc38a2*), while two others (*Gnao1*—*Gng8*, *Izts3*—*Prkcg*) were switched from synergistically expressed in males to antagonistically expressed in females. Moreover, as shown in panels (b) and (c), several independently expressed gene pairs in one sex became significantly synergistically/antagonistically expressed in the other. All these differences indicate distinct molecular mechanisms involved in the glutamatergic synaptic transcription.

### 3.8. Sex Dichotomy of the Genes’ Hierarchy

Table 4 lists the most influential neurotransmission genes (higher GCH) and the Gene Master Regulators in all six groups of the profiled samples. The relevant neurotransmission genes, *Gnb4*, *Gng10*, and *Gng12,* are part of all five synaptic pathways. *GCHs* were computed using the software #GENE COMMANDING HEIGHT# [52]. Of note is the substantially lower GCH scores of the top neurotransmission genes with respect to the corresponding GMRs in the six groups: *Erg28* (40 vs. 20 for *Pik3r2* in MSN), *Erap1* (70 vs. 26 for *Abat* in FSN), *Tmem238* (162 vs. 16 for *Gphn* in MBN), *Taf8* (38 vs. 14 for *Grm8* in FBN), and *Tmem134* (78 vs. 19 for *Mapk10* in MBY, respectively, 78 vs. Homer1 in F-BYS).

## 4. Discussion

While a certain sex-specific prevalence of IESS in boys was discovered in large cohort studies [22], and we also saw trends to increased susceptibility in males, this never reached statistical significance in our relatively small experimental cohorts (<20 per sex group) on an IESS rat model. Nevertheless, the very sensitive and unbiased transcriptomic analyses were able to provide a glimpse of the significant sex dichotomy in the genomic molecular machinery.

The analyses of six neurotransmission pathways in the PVNs of male and female rats at P16 revealed substantial transcriptomic differences between the two sexes, which extend, although at different levels, to the betamethasone prenatally primed pups with and without NMDA-triggered spasms.

Beyond traditional gene expression studies that are limited to quantifying the expression profile, our approach incorporates two additional independent measures of the individual genes: the control of transcripts’ abundances and the inter-coordination of their expression. Together, the three independent measures provide the most comprehensive characterization of the two sexes’ PVN transcriptomic topologies and their differential remodeling in infantile spasms.

Although the tissue pieces were very small, they were still heterocellular, which is one major limitation of this study. In the worst scenario, the non-significant change in genes, when comparing different conditions for the same sex or different sexes for the same condition, might result from up-regulation in a particular phenotype and down-regulation in another. However, taking a particular type of cell from its natural environment would have a larger effect on the transcriptome, as we proved by profiling cortical astrocytes and precursor oligodendrocytes when co-cultured and cultured separately [55,56,57].

We found (Table 1) few sex differences between the expression levels of the top neurotransmission genes. For instance, the active regulator of intracellular Ca^2+^ release, *Caly,* has the largest expression level among all neurotransmission genes in the PVNs of both sexes in all three investigated conditions. An abundance of *Cali* transcripts was between 126 and 195 times larger than the expression level of the median gene in the respective group, close enough to *Cst3* and *Rpl41*, the top-expressed genes in the entire transcriptome. *Cali*, localized in the neuron dendritic spines, is related to D1 dopaminergic transmission and schizophrenia [31,32]. However, the expression of *Caly* (Table 2) was less controlled than the median gene in all groups, *REC* = −0.98 in MSN, −0.41 in FSN, −1.26 in MBN, −0.90 in FBN, −0.85 in MBY, and −0.69 in FBY, indicating remarkable flexibility.

As seen in Table 2, the most controlled neurotransmission genes are as follows: *Pik3r2* and *Abat* in the two SN groups, *Pik3r5* and *Grm8* in BNSs, and *Th* and *Trpc1* in BYSs. Interestingly, a mutation of *Pik3R2* was associated with familial temporal lobe epilepsy [58], and its overexpression may reduce cell viability and boost autophagy and apoptosis [59]. GABA transaminase deficiency caused by a mutation of *Abat* leads to neonatal epilepsy [60], while activation through the FOXA2/ABAT/GABA axis mediates the development of brain metastasis in lung cancer [61]. Therefore, the high control of these two genes in the normal condition (SN) prevents neurotransmission alterations associated with IEES in the corresponding sex.

Moreover (see Appendix A for the M/F ratios of RCS), neurogenes like *Mapk10* in SN, *Pik3r5* (BN), and *Th* (BY) are strictly controlled in males but allowed to fluctuate in females in the corresponding conditions. In contrast, genes like *Abat* and *Gabbr1* in SN and *Trpc1* (BY) are flexibly expressed in males but strictly controlled in females in the corresponding conditions. Both *Mapk10* and *Gabbr1* are considered potential targets for vascular dementia treatment [62,63]. The differences are even larger for the top controlled genes in each condition, with *Erg28*, *Tmem238,* and *Oxsr1* very strictly controlled in males but much less controlled in females, while the opposite is observed for *Erap1*, *Cul7,* and *Tac2.* All these differences point to distinct homeostatic mechanisms that control the transcripts’ abundances.

The effect of IEES triggering in betamethasone primed rats (BY condition) has distinct sex-dependent effects on genes’ RCSs (Appendix A). While the RCSs of *Pik3r2* are increased by IEES in males (by 6.49×), with little effect on females, that of *Gnal* is increased in females (by 8.84×). *Gnal* having greater effects on female than on male dystonia was recently documented in a population study [64].

There are also substantial sex differences among the most flexibly expressed neuro-transmission genes (Table 3) in all three conditions: *Calml1* vs. *Mapk10* in SN, *Drd2* vs. *Creb3* in BN, and *Clock* vs. *Grin1* in BY. Interestingly, all the top flexible genes are included in the dopaminergic synapse pathway. Although several other reports discussed sex differences in the expressions of neurotransmission genes and their encoded proteins (e.g., [65,66,67,68,69]), here, we report, for the first time, the sex dichotomy in gene expression control.

*Th*, a rate-limiting enzyme (tyrosine hydroxylase) in dopamine, epinephrine, and norepinephrine biosynthesis [70], has a spectacular (by 44.51×) RCS increase in IESS with respect to the healthy counterpart in males (Appendix A), which is also 14.58× larger than in females’ BYSs. Therefore, we consider that *Th* might be a potential gene therapy target for future research for a male with IESS but not for a female with IESS.

In contrast, the control strength of *Creb3* in females decreased significantly (by 8.68×) in both BN and BY conditions (Appendix A). The *Creb3* protein tethers chromatin to the cell’s inner nuclear membrane and prevents karyoptosis, a type of cell death caused by DNA release into the cytosol [71]. Therefore, the high expression flexibility of the encoding gene makes both cholinergic and dopaminergic transmission more adaptable to prenatal corticosteroids, even in the absence of IS.

Figure 2a shows a higher synaptic vesicle acidification following endocytosis but a lower neurotransmission uptake in male versus female PVNs. Apparently, these differences indicate that in males, there is a higher efficiency in recycling the synaptic vesicles, yet a decreased release of neurotransmitters compared to females. However, as demonstrated in Figure 3a,b, IESS induction does not cause an overall imbalance of the synaptic vesicle cycle in either sex. Interestingly, reports by others demonstrated that some factors, such as exposure to diazepam [72] or maternal immune activation [73], affect the synaptic vesicle cycle pathway. We further found substantial differences between the significantly regulated genes by IESS in males (Figure 2a) and in females (Figure 2b). For instance, while in males, the ATPase *Atp2a2* (also known as. *Serca2*), is involved in actively pumping Ca^2+^ from the cytosol into the endoplasmic reticulum and is a candidate gene for IESS [74] is down-regulated, it is up-regulated in females. We can speculate that this calcium homeostasis regulator [75] may be responding to different cytosolic calcium conditions after betamethasone/spasm exposure in males and females. While *ATP2A2* is a monogenic cause of Darier disease (a skin disorder with neuropsychiatric abnormalities), the prevalence of epilepsy in this condition is higher than in the general population [76]. Interestingly, our unpublished in vitro data (not stratified for sex) indicate that hippocampal LTP is diminished after prenatal betamethasone exposure, and this effect is independent of NMDA-induced spasms, and increasing the number of subjects and stratification by sex may fully reveal the biological importance of this finding.

Figure 2b indicates lower glutamatergic transmission in control saline-injected (i.e., SN) males compared to the female PVNs, caused by the underexpression of *Grik1* and Slc1a2 in the presynaptic neuron. The higher expression of *Grik1* in women relative to that in men was also detected in patients with depression [77]. Through the underexpression of *Gabra4*, *Gabrd*, and *Gabrdg1*, Figure 2c confirms the report [78] of male rats lagging behind females in the development of the ionotropic of GABA-A receptors. The underexpression of the G proteins *Gnb4* and *Gng5* in males with respect to their female counterparts was common to all five investigated synapse pathways (Figure 2b–f). These G proteins are involved in the presynaptic inhibition, diminishing the release of glutamate, GABA, and Ach release in the synaptic cleft [79,80]. However, the effect of the reduced expression of the G proteins is compensated by the increased expression of the inositol 1,4,5-trisphosphate receptors *Itpr1* and *Itpr2* (*Itpr3* was not quantified). The activation of these receptors releases Ca^2+^ (which controls almost all important cellular processes [81]) from the intracellular IP_3_-sensitive storage [82].

The differences between the two sexes also appear in the regulomes of SVC (Figure 3), GLU (Figure 4), and GABA (Figure 5) pathways after the induction of IESS. For instance, the presynaptic synaptic vesicle regulatory protein *Cplx2* (complexin 2), whose variants affect cognition and memory in schizophrenic patients [83,84], is down-regulated in males (Figure 3a) but up-regulated in females (Figure 3b). A very recent study looking at the percentage reads of new isoforms identified new *CPLX2* isoforms not only in patients with schizophrenia but also in patients with epilepsy [85]. Our finding of altered *Cplx2* expression indirectly corroborates a study on chronic mild stress that links *Cplx2* to the anxiety-susceptible experimental rat group [86]. The expression of several genes (e.g., Kcnj3 and *Trpc1*) was not affected in one sex (female) but was significantly regulated in the other. These findings suggest that dissimilar pathological processes affected the neurotransmission in the two sexes. Among others, our result explains why the suppression of the potassium channel encoded by *Kcnj3* impairs prelimbic cortical function in male but not female mice [87].

Figure 6 presents the spectacular differences between the male and female statistically significant SVC pathway transcriptome networks. Thus, three antagonistically expressed gene pairs in males (*Atp6v0a2–Atp6v0a4*, *Atp6v1g2–Dnm2*, *Dnm2-Nsf*) were switched to synergistically expressed pairs in females, while the pair *Cacna1a–Slc6a7* was switched from synergistically expressed in males to antagonistically expressed in females (Figure 6a). Moreover, six independently expressed gene pairs in males were significantly synergistically or antagonistically correlated in females (Figure 6b), and eight synergistically or antagonistically expressed in males were independently expressed in females (Figure 6c).

Given the roles of v-ATPase subunits in phagocytosis, endocytosis, and autophagy [88], it would be interesting to study the functional consequences of the opposite expression coordination of *Atp6v0a2* and *Atp6v0a4* in the two sexes. Of note is the sex discrepancy in the expression correlation of the dynamin *Dnm2*, responsible for vesicle recovery after releasing the neurotransmitters into the synaptic cleft, with the v-ATPase *Atp6v1g2* involved in vesicle acidification needed for neurotransmitter uptake. Also surprising is the antagonistic expression of *Nsf* (which removes the cis-SNARE complex [89]) and *Dnm2* in males, while they are supposed to stimulate each other’s expression (as it happens in females).

Likewise, we found substantial sex differences in the transcriptomic networks of the GLU pathway. Thus, the antagonistically expressed pairs in males, *Gng13-Gls2*, *Gria2-Gnao1*, *Itpr1-Gng5*, and *Slc38a2-Grm2,* are synergistically expressed in females, while the synergistically expressed pairs in males, *Gng8-Gnao1*, and *Prkcg-Lzts3,* are antagonistically expressed in females (Figure 6a). In addition, six significantly correlated gene pairs in males are independently expressed in females, and 20 (*sic!*) independently expressed pairs in males are significantly correlated in females. The opposite expression correlations of *Gng13* with *Gls2* in the two sexes suggest an opposite relationship between mitophagy and the feedback inhibition of glutamate release. Such coordination might have consequences on the epilepsy occurrence [90], where *Gls2* is down-regulated (as we also found, Figure 4a).

All these transcriptomic network differences indicate distinct molecular mechanisms responsible for the formation of synaptic vesicles, the release of neurotransmitters, and the response by the post-synaptic neurons in the PVNs of the two sexes. The neurotransmission differences are most likely responsible for different brain circuitries in males and females (e.g., [91,92,93]).

Interestingly, we observed that the genes involved in neurotransmission are not among the most influential in both male and female rats subjected to each of the three conditions. Yet, the most prominent genes in the IESS condition, *Grm8* (male) and *Mapk10* (female), have documented implications in epilepsy (e.g., [94,95,96]). However, the present study pinpoints *Tmem134*, a protein located in the perinuclear region of the cytoplasm, involved in RNA splicing [97] and associated with obesity [98], as the best target for the gene therapy of IESS in rats of both sexes based on symmetrical expression change in both sexes. Additionally, our model suggests a significant involvement of hypothalamic nuclei, such as the arcuate nucleus [7], also participating in the control of food intake. *Tmem134* is also involved in the regulation of the NF-kappaB pathway [33]. Activity in this pathway is increased after seizures, and the down-regulation of this pathway attenuates seizures [99]. Hence, affecting this pathway through Tmem134 may provide a feasible way for the control of spasms. In previous reports, we assumed and verified [52,100] that the manipulation of genes with higher *GCHs* has greater consequences on the transcriptome, the top gene, named the Gene Master Regulator (GMR), being the most influential, whose silence might be lethal for the cell.

There are several limitations of this study. First, a sample size of 12 is too low to provide reliable stratification by sex, especially if biological variables such as spasms or body weight are considered. Second, our transcriptomic data were not further validated in a proteomic experiment. This may discount the observed effects only to the transcriptome without translation to the effector proteins. Since, in addition to translation, protein abundance is affected by several post-translational modifications, and the genes’ expression correlations do not exactly overlap the protein–protein interactions, our results are relevant only at the transcriptomic level, and integration with epigenetic and proteomic data is needed to strengthen the conclusion. However, our electrophysiology data [101], as well as a behavioral study [102], indicate electrophysiological as well as behavioral effects of prenatal betamethasone exposure, though larger sample sizes would be required for proper stratification by sex. Finally, the effects of repetitive spasms during development (P12, P13, P15) on behaviors and delayed seizure susceptibility are still being investigated and therefore are missing as comparators.

## 5. Conclusions

Our transcriptomic analysis of six neurotransmission pathways in the PVNs of P16 12 male and 12 female rats revealed substantial sex differences, persisting even in prenatally betamethasone-primed pups, regardless of NMDA triggering of IESS. By integrating expression level, transcript abundance control, and expression inter-coordination, we provide the most theoretically possible comprehensive transcriptomic characterization of these differences. Nonetheless, because transcript abundances and protein content are not proportional [103,104] (gene transcription is triggered by the necessity to maintain a certain level of proteins), our results cannot be automatically translated into sex-dependent proteome topology and remodeling by IESS.

Sex-specific transcriptomic shifts in synaptic vesicle cycling, glutamatergic, and GABAergic pathways suggest distinct pathological mechanisms, influenced by the sex hormones [105,106,107]. We found that gene co-expression patterns differed, highlighting the fundamental sex-dependent synaptic organization that determines brain circuits (e.g., [108,109,110]). This study extends our previous findings of sex transcriptomic dichotomy in regions of the brain [8,9,10], heart [111,112], and kidneys [113].

Finally, in our rat model, *Tmem134* (encoding cytosolic and membrane proteins likely involved in the cytokine pathway [33]) emerged as the most influential gene for IESS pathology in both sexes. Nevertheless, the dominance of *Tmem134* is surprising and deserves to be tested in further experiments on the same IEES rat model.

## Figures and Tables

**Figure 1 cimb-47-00380-f001:**
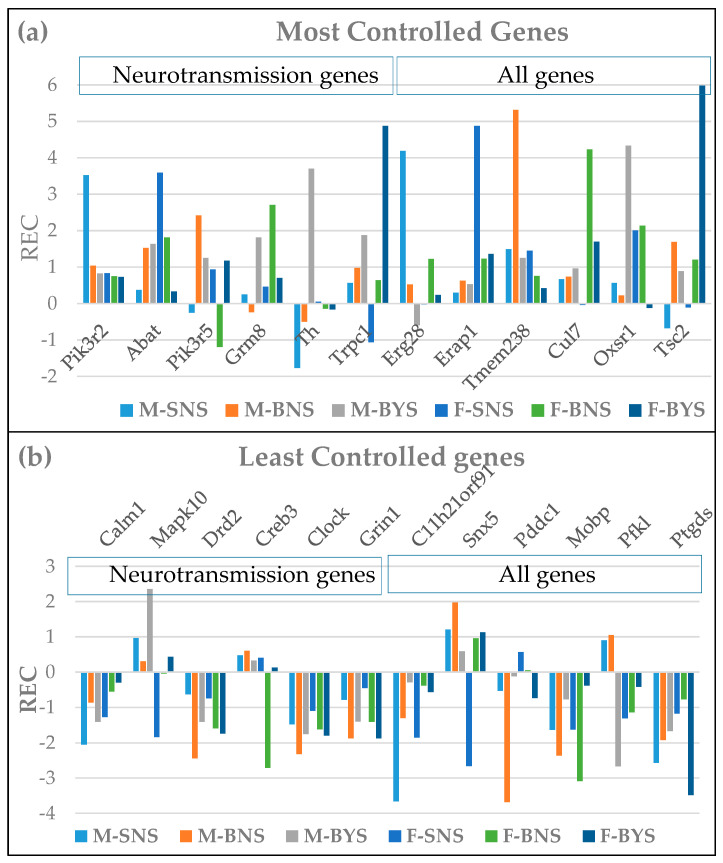
Relative Expression Control (*REC*) of the extremely controlled and extremely flexibly expressed neurotransmission genes in comparison with all other genes in the male and female paraventricular nuclei of rats with/without betamethasone priming, with/without ISEE NMDA triggering. (**a**) Top controlled (highest REC). (**b**) Most flexibly expressed genes. Note the substantial differences among the three conditions in each sex as well as between the two sexes in each condition.

**Figure 2 cimb-47-00380-f002:**
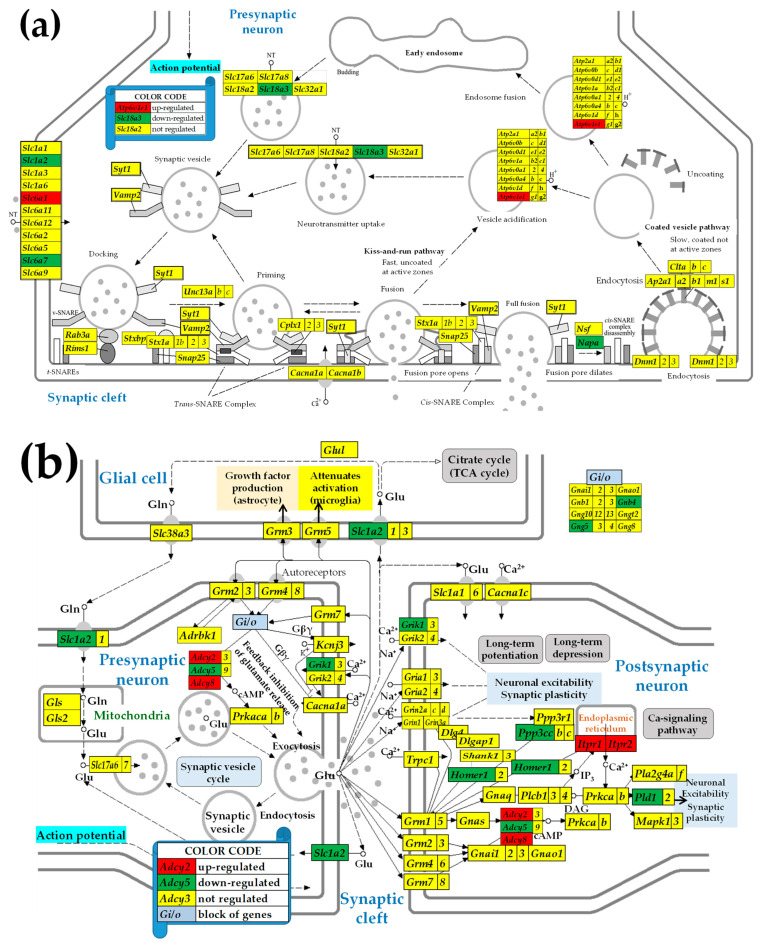
Sex differences in the unaltered (SN) state of the KEGG-constructed pathways: (**a**) synaptic vesicle cycle, (**b**) Glutamatergic Synapse, (**c**) GABAergic, (**d**) cholinergic, (**e**) dopaminergic, and (**f**) serotonergic synapses in the PVNs of males and females. The red/green background of the gene symbols indicates whether that gene was significantly over/underexpressed in the males with respect to the females’ tissue, while the yellow background indicates that the expression difference was not significant. Differentially expressed genes: *Abat* (4-aminobutyrate aminotransferase), *Adcy2/5/8* (adenylate cyclase 2/5/8), *Akt1* (v-akt murine thymoma viral oncogene homolog 1), *Atp6v1c1* (ATPase, H+ transporting, lysosomal V1 subunit C1), *Chrnb4* (cholinergic receptor, nicotinic, beta 4 (neuronal)), *Gabra4/d/g1* (gamma-aminobutyric acid (GABA) A receptor, alpha 4/delta/gamma1), *Gnb4* (guanine nucleotide-binding protein (G protein), beta polypeptide 4), *Gng5* (guanine nucleotide-binding protein (G protein), gamma 5), *Grik1* (glutamate receptor, ionotropic, kainate 1), *Homer1* (homer homolog 1 (Drosophila)), *Itpr1/2* (inositol 1,4,5-trisphosphate receptor, type 1/2), *Kcnj14* (potassium inwardly rectifying channel, subfamily J, member 14), *Kras* (Kirsten rat sarcoma viral oncogene), *Mapk11* (mitogen-activated protein kinase 11), *Napa* (N-ethylmaleimide-sensitive factor attachment protein, alpha), *Pik3ca* (phosphoinositide-3-kinase, catalytic, alpha polypeptide), *Pld1* (phospholipase D1), *Ppp1ca* (protein phosphatase 1, catalytic subunit, alpha isozyme), *Ppp3cc* (protein phosphatase 3, catalytic subunit, gamma isozyme), *Slc18a3* (solute carrier family 18 (vesicular acetylcholine transporter), member 3), *Rapgef3* (Rap guanine nucleotide exchange factor (GEF) 3), *Slc1a2* (solute carrier family 1 (glial high-affinity glutamate transporter), member 2), *Slc18a3* (solute carrier family 18 (vesicular acetylcholine transporter), member 3, and *Slc6a1/7* (solute carrier family 6 (neurotransmitter transporter), member 1/7).

**Figure 3 cimb-47-00380-f003:**
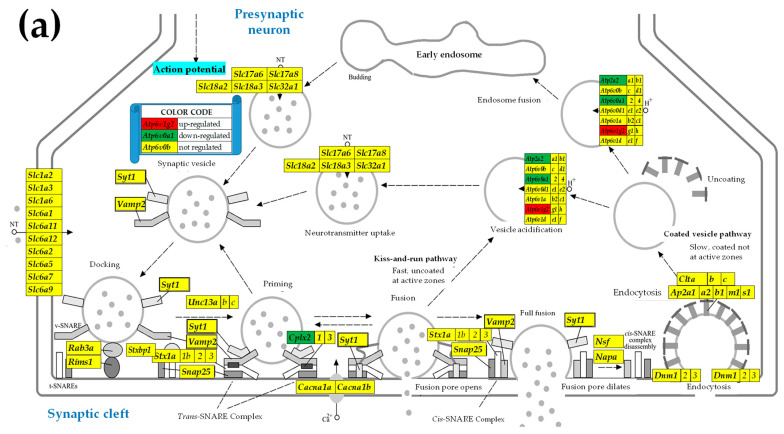
Sex differences in the regulome of the KEGG-constructed synaptic vesicle cycle (SVC) of the betamethasone-primed PVN following the induction of the spasms. (**a**) Male rats. (**b**) Female rats. The red/green background of the gene symbols indicates whether that gene was significantly over/underexpressed in the males with respect to the females’ tissue, while the yellow background indicates that the expression difference was not significant. Differentially expressed genes: *Ap2a2* (adaptor-related protein complex 2, alpha 2 subunit), *Atp2a2* (ATPase, Ca++ transporting, cardiac muscle, slow twitch 2), *Atp6v0c* (ATPase, H+ transporting, lysosomal V0 subunit C), *Atp6v1c1/g2*, *Cplx2* (complexin 2), and *Slc6a1*.

**Figure 4 cimb-47-00380-f004:**
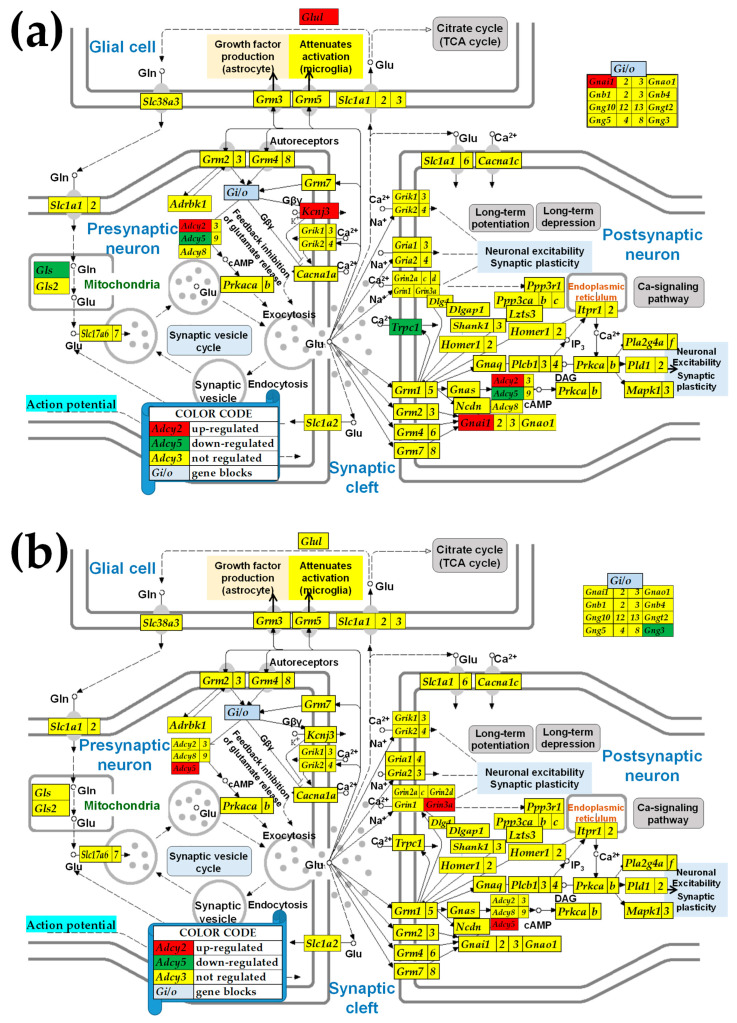
Significantly regulated genes in the KEGG-constructed Glutamatergic Synapse (GLU) pathway of the betamethasone-primed PVNs of male (**a**) and female (**b**) rats following the induction of the infantile spasms. The red/green background of the gene symbols indicates whether that gene was significantly up/down-regulated by the induction of spasms in the betamethasone-primed animals, while the yellow background indicates not significant regulation. Regulated genes: *Adcy2/5*, *Gls* (glutaminase (Gls), nuclear gene encoding mitochondrial protein), *Glul* (glutamate-ammonia ligase)*, Gnai1* (guanine nucleotide-binding protein (G protein), alpha inhibiting activity polypeptide 1)*, Kcnj3* (potassium inwardly rectifying channel, subfamily J, member 3), and *Trpc1* (transient receptor potential cation channel, subfamily C, member 1).

**Figure 5 cimb-47-00380-f005:**
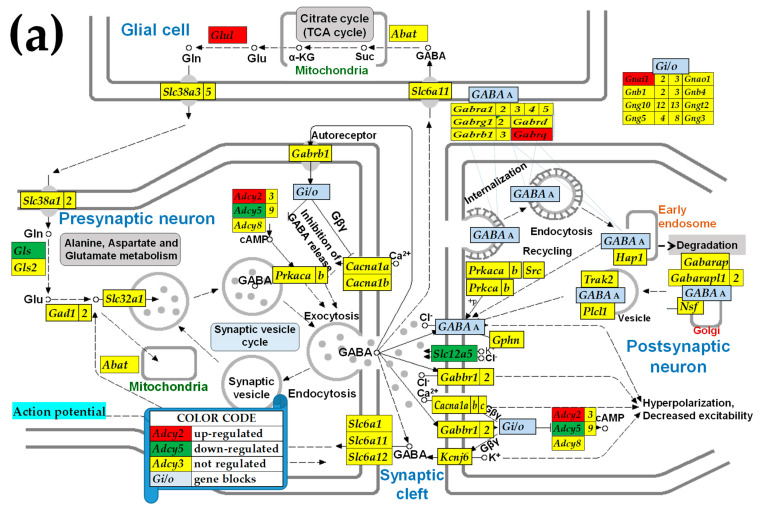
Significantly regulated genes in the KEGG-constructed GABA synapse (GABA) pathway of the betamethasone-primed PVNs of male (**a**) and female (**b**) rats following the induction of the infantile spasms. The red/green background of the gene symbols indicates whether that gene was significantly up/down-regulated by the induction of spasms in the betamethasone-primed animals, while the yellow background indicates not significant regulation. Regulated genes: *Adcy2/5*, *Gabra1*, *Gabrq*, *Gabarapl2* (GABA(A) receptor-associated protein like 2), *Gls*, *Glul*, *Gnai1*, *Gng3*, *Slc12a5*, and *Slc6a1* (solute carrier family 6 (neurotransmitter transporter), member 1).

**Figure 6 cimb-47-00380-f006:**
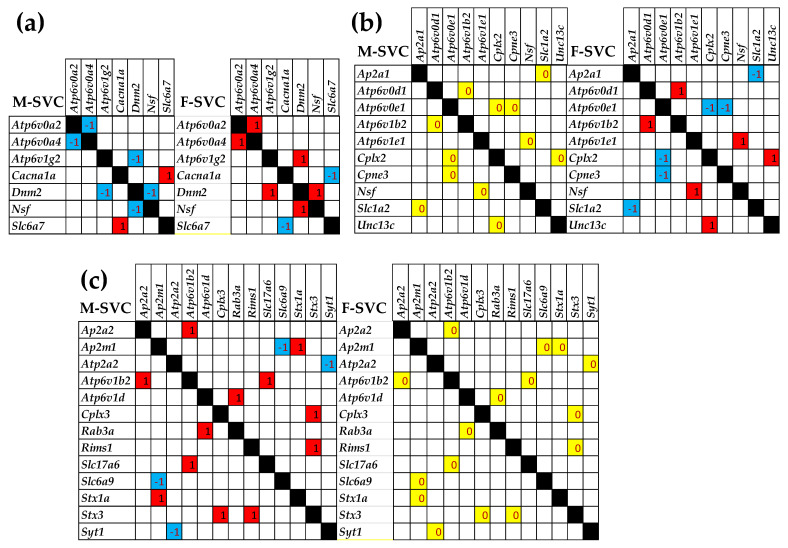
Significant sex dichotomy in the expression correlations among the SVC genes. (**a**) Gene pairs that switch their significant synergistic (red square) or antagonistic (blue square) correlation from male to the opposite in female. (**b**) Independently expressed gene pairs (yellow square) in males that became significantly synergistically (red square) or antagonistically (blue square) correlated in females. (**c**) Significantly correlated gene pairs in males that are independently expressed in females. Interesting genes: *Ap2a1/m1* (adaptor-related protein complex 2, alpha 1/m1 subunit), *Atp2a2* (ATPase, Ca++ transporting, cardiac muscle, slow twitch 2), *Atpv0a2/4* (ATPase, H+ transporting, lysosomal V0 subunit A2/4), *Atp6v0d1/e1* (ATPase, H+ transporting, lysosomal V0 subunit D1/e1), *Atp6v1b2/e1/g2* (ATPase, H+ transporting, lysosomal V1 subunit B2/E1/G2), *Cacna1a* (calcium channel, voltage-dependent, P/Q type, alpha 1A subunit), *Cplx2* (complexin 2), *Cpne3* (copine III), *Dnm2* (dynamin 2), *Nsf* (N-ethylmaleimide-sensitive factor), *Rab3a* (RAB3A, member RAS oncogene family), *Slc1a2* (solute carrier family 1 (glial high-affinity glutamate transporter)), and *Unc13c* (unc-13 homolog C).

**Figure 7 cimb-47-00380-f007:**
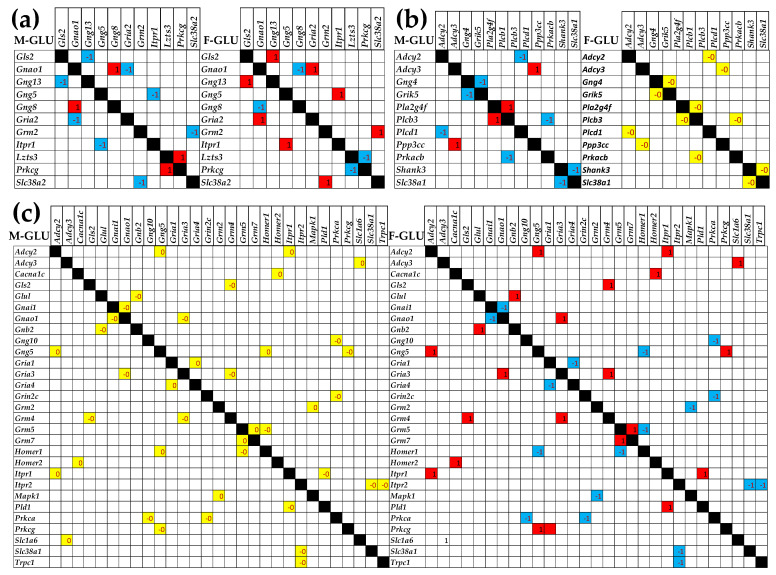
Significant sex dichotomy in the expression correlations among the GLU genes. (**a**) Gene pairs that switch their significant synergistic (red square) or antagonistic (blue square) correlation in one sex to the opposite in the other. (**b**) Significantly synergistically (red square) and antagonistically (blue square) correlated gene pairs in males that were independently (yellow square) expressed in females. (**c**) Significantly independently (yellow square) expressed genes in males that were significantly synergistically (red square) or antagonistically (blue square) expressed in females.

**Table 1 cimb-47-00380-t001:** The top five most expressed neurotransmission genes in the PVNs of the two sexes subjected to each of the three conditions. Numbers in the MALE and FEMALE SN, BN, and BY columns are the average expression levels normalized to the expression level of the median gene in that condition for each sex. For instance, the expression of *Caly* in male SN is 126 times larger than that of the median gene in the PVNs of the saline-primed rats without infantile spasms. Note the similarity of the gene hierarchy in each group of samples.

	Top 5 Expressed Neurotransmission Genes		Male			Female		
Gene	Description	PATH	SN	BN	BY	SN	BN	BY
*Caly*	calcyon neuron-specific vesicular protein	4	126	139	126	195	143	155
*Gabarapl1*	GABA(A) receptor-associated protein like 1	2	104	112	80	69	108	129
*Ap2m1*	adaptor-related protein complex 2, mu 1 subunit	0	96	110	89	125	112	122
*Calm2*	calmodulin 2	4	72	70	60	60	75	72
*Hap1*	huntingtin-associated protein 1	2	63	63	67	73	72	73
*Caly*	calcyon neuron-specific vesicular protein	4	126	139	126	195	143	155
*Ap2m1*	adaptor-related protein complex 2, mu 1 subunit	0	96	110	89	125	112	122
*Mapk3*	mitogen activated protein kinase 3	135	59	64	67	85	68	68
*Gnai2*	guanine nucleotide binding protein (G protein), alpha inhibiting activity polypeptide 2	12345	57	64	57	75	64	74
*Hap1*	huntingtin-associated protein 1	2	63	63	67	73	72	73
*Caly*	calcyon neuron-specific vesicular protein	4	126	139	126	195	143	155
*Gabarapl1*	GABA(A) receptor-associated protein like 1	2	104	112	80	69	108	129
*Ap2m1*	adaptor-related protein complex 2, mu 1 subunit	0	96	110	89	125	112	122
*Calm2*	calmodulin 2	4	72	70	60	60	75	72
*Gnai2*	guanine nucleotide binding protein (G protein), alpha inhibiting activity polypeptide 2	12345	57	64	57	75	64	74
*Caly*	calcyon neuron-specific vesicular protein	4	126	139	126	195	143	155
*Ap2m1*	adaptor-related protein complex 2, mu 1 subunit	0	96	110	89	125	112	122
*Gabarapl1*	GABA(A) receptor-associated protein like 1	2	104	112	80	69	108	129
*Calm2*	calmodulin 2	4	72	70	60	60	75	72
*Hap1*	huntingtin-associated protein 1	2	63	63	67	73	72	73
*Caly*	calcyon neuron-specific vesicular protein	4	126	139	126	195	143	155
*Ap2m1*	adaptor-related protein complex 2, mu 1 subunit	0	96	110	89	125	112	122
*Gabarapl1*	GABA(A) receptor-associated protein like 1	2	104	112	80	69	108	129
*Mapk3*	mitogen activated protein kinase 3	135	59	64	67	85	68	68
*Hap1*	huntingtin-associated protein 1	2	63	63	67	73	72	73
*Caly*	calcyon neuron-specific vesicular protein	4	126	139	126	195	143	155
*Gabarapl1*	GABA(A) receptor-associated protein like 1	2	104	112	80	69	108	129
*Ap2m1*	adaptor-related protein complex 2, mu 1 subunit	0	96	110	89	125	112	122
*Gnai2*	guanine nucleotide binding protein (G protein), alpha inhibiting activity polypeptide 2	12345	57	64	57	75	64	74
*Hap1*	huntingtin-associated protein 1	2	63	63	67	73	72	73

**Table 2 cimb-47-00380-t002:** The most controlled neurotransmission genes in the PVNs of the two sexes subjected to each of the three conditions. Numbers in the MALE and FEMALE SN, BN, and BY columns are the Relative Expression Control (*REC*) scores. Note the alteration of the normal hierarchy of controlled genes in (M/F)SN and (M/F)BY conditions and the substantial differences between the two sexes in all three conditions. For comparison, the table includes the *REC*s of the most controlled non-neurotransmission genes (no number in the “Path” column): *Erg88*, *Erap1*, *Tmem238*, *Cul7*, *Oxsr1*, and *Tsc2*.

	Top 5 Most Controlled Neurotransmission Genes		Male			Female		
Gene	Description	PATH	SN	BN	BY	SN	BN	BY
*Pik3r2*	phosphoinositide-3-kinase, regulatory subunit 2 (beta)	3	3.52	1.04	0.82	0.83	0.75	0.73
*Atp6v0b*	ATPase, H+ transporting, lysosomal V0 subunit B	0	2.40	−0.59	1.22	1.01	0.12	−0.02
*Calml4*	calmodulin-like 4	4	2.15	−0.61	0.65	−0.56	−0.02	−0.71
*Ppp1ca*	protein phosphatase 1, catalytic subunit, alpha isozyme	4	2.13	0.55	1.12	1.16	2.00	1.32
*Atp6v1g2*	ATPase, H+ transporting, lysosomal V1 subunit G2	0	1.95	−0.32	0.67	0.06	0.87	1.39
*Abat*	4-aminobutyrate aminotransferase	2	0.38	1.52	1.63	3.58	1.81	0.33
*Gabbr1*	gamma-aminobutyric acid (GABA) B receptor 1	2	−0.26	0.72	0.75	2.87	0.15	1.23
*Slc6a7*	solute carrier family 6 (neurotransmitter transporter), member 7	0	1.08	−0.01	1.01	2.82	−0.08	1.63
*Chrnb4*	cholinergic receptor, nicotinic, beta 4 (neuronal)	3	0.87	0.09	0.40	2.75	0.60	0.11
*Gnal*	GTP-binding protein Golf alpha subunit	4	0.83	−0.98	−0.45	2.66	−0.28	−0.48
*Pik3r5*	phosphoinositide-3-kinase, regulatory subunit 5	3	−0.25	2.42	1.25	0.94	−1.19	1.17
*Atp2a2*	ATPase, Ca++ transporting, cardiac muscle, slow twitch 2	0	0.60	2.18	0.81	−0.60	1.53	0.94
*Ppp2r5e*	protein phosphatase 2, regulatory subunit B′, epsilon isoform	4	0.76	2.04	2.08	−0.44	0.85	0.47
*Ppp2r3c*	protein phosphatase 2, regulatory subunit B″, gamma	4	1.70	1.85	0.72	−0.32	0.10	1.20
*Slc38a5*	solute carrier family 38, member 5	2	1.94	1.81	0.50	−0.09	−0.88	0.94
*Grm8*	glutamate receptor, metabotropic 8	1	0.25	−0.24	1.81	0.47	2.70	0.70
*Cpne3*	copine III	0	0.23	0.68	0.58	−0.98	2.39	−0.20
*Atp6v1h*	ATPase, H+ transporting, lysosomal V1 subunit H	0	0.80	1.55	0.50	−0.24	2.30	0.57
*Gls*	glutaminase (Gls), nuclear gene encoding mitochondrial protein	12	0.04	1.07	1.27	−1.34	2.19	0.76
*Gabarapl2*	GABA(A) receptor-associated protein like 2	2	0.66	0.62	1.10	−1.40	2.09	1.43
*Th*	tyrosine hydroxylase	4	−1.77	−0.50	3.71	0.05	−0.14	−0.16
*Maoa*	monoamine oxidase A	45	0.59	0.53	2.62	1.17	0.93	−0.42
*Mapk10*	mitogen activated protein kinase 10	4	0.97	0.30	2.35	−1.84	−0.04	0.44
*Scn1a*	sodium channel, voltage-gated, type I, alpha	4	−0.30	0.69	2.29	1.00	−0.67	0.30
*Gabrg1*	gamma-aminobutyric acid (GABA) A receptor, gamma 1	2	0.24	0.04	2.22	0.79	1.76	−0.87
*Trpc1*	transient receptor potential cation channel, subfamily C, member 1	15	0.57	0.98	1.87	−1.06	0.64	4.87
*Gng10*	guanine nucleotide binding protein (G protein), gamma 10	12345	0.56	1.08	0.11	0.05	1.38	2.79
*Gng8*	guanine nucleotide binding protein (G protein), gamma 8	1245	0.20	0.37	0.40	0.78	−0.18	2.58
*Raf1*	v-raf-leukemia viral oncogene 1	5	1.05	1.68	0.68	0.98	−0.64	2.21
*Atp6v0a1*	ATPase, H+ transporting, lysosomal V0 subunit A1	0	−0.21	0.11	1.68	−0.69	2.06	2.04

**Table 3 cimb-47-00380-t003:** The least controlled neurotransmission genes in the PVNs of the two sexes subjected to all three conditions. Numbers in the MALE and FEMALE SN, BN, and BY columns are the Relative Expression Control (*REC*) scores. Note the alteration of the normal hierarchy of controlled genes in the BNS and BYS conditions and the substantial differences between the two sexes in all three conditions. For comparison, the table also includes the least controlled non-neurotransmission genes in each group of samples: *C11h21orf91*, *Snx5*, *Pddc1*, *Mobp*, *Pfk1*, and *Ptgds*.

	Top 5 Least Controlled Neurotransmission Genes		Male			Female		
Gene	Description	PATH	SN	BN	BY	SN	BN	BY
*Calm1*	calmodulin 1	4	−2.05	−0.87	−1.41	−1.27	−0.56	−0.30
*Alox15*	arachidonate 15-lipoxygenase	5	−2.04	−0.40	−1.20	−0.03	−1.23	−1.47
*Th*	tyrosine hydroxylase	4	−1.77	−0.50	3.71	0.05	−0.14	−0.16
*Plcb1*	phospholipase C, beta 1 (phosphoinositide-specific)	1345	−1.73	−1.61	−1.06	−0.27	−1.28	−1.58
*Clock*	clock circadian regulator	4	−1.48	−2.33	−1.76	−1.10	−1.61	−1.80
*Mapk10*	mitogen activated protein kinase 10	4	0.97	0.30	2.35	−1.84	−0.04	0.44
*Gna11*	guanine nucleotide binding protein, alpha 11	3	−0.54	−1.47	−1.16	−1.76	−0.62	−1.41
*Gabarapl1*	GABA(A) receptor-associated protein like 1	2	−0.72	−0.64	−0.90	−1.74	−0.66	−0.53
*Rims1*	regulating synaptic membrane exocytosis 1	0	−1.17	−1.73	−1.37	−1.69	−1.41	−1.67
*Map2k1*	mitogen activated protein kinase kinase 1	35	−0.51	0.19	0.57	−1.64	1.03	−0.19
*Drd2*	dopamine receptor D2	4	−0.63	−2.45	−1.42	−0.74	−1.59	−1.74
*Slc6a12*	solute carrier family 6 (neurotransmitter transporter), member 12	02	−0.58	−2.35	−1.35	−0.61	−1.29	−1.62
*Clock*	clock circadian regulator	4	−1.48	−2.33	−1.76	−1.10	−1.61	−1.80
*Alox12b*	arachidonate 12-lipoxygenase, 12R type	5	−1.16	−2.25	−1.39	−0.27	−1.53	−1.66
*Gnb3*	guanine nucleotide binding protein (G protein), beta polypeptide 3	12345	−0.88	−2.24	−1.57	−0.96	−1.56	−1.81
*Creb3*	cAMP responsive element binding protein 3	34	0.47	0.61	0.33	0.41	−2.71	0.13
*Gng5*	guanine nucleotide binding protein (G protein), gamma 5	12345	0.65	0.52	1.23	0.40	−2.22	1.47
*Camk2b*	calcium/calmodulin-dependent protein kinase II beta	34	0.90	−0.60	0.39	−1.08	−2.15	−0.05
*Akt1*	v-akt murine thymoma viral oncogene homolog 1	34	0.46	−0.23	0.29	−0.46	−2.15	−0.87
*Slc1a3*	solute carrier family 1 (glial high affinity glutamate transporter), member 3	01	−0.23	−0.48	−0.77	0.23	−2.07	−0.17
*Clock*	clock circadian regulator	4	−1.48	−2.33	−1.76	−1.10	−1.61	−1.80
*Gria2*	glutamate receptor, ionotropic, AMPA 2	14	0.28	−0.51	−1.64	−1.18	0.77	−0.32
*Htr7*	5-hydroxytryptamine (serotonin) receptor 7, adenylate cyclase-coupled	5	−0.94	−2.17	−1.60	−0.78	−1.66	−1.71
*Unc13b*	unc-13 homolog B (C. elegans)	0	−1.07	−2.22	−1.60	−0.24	−1.63	−1.78
*Akt2*	v-akt murine thymoma viral oncogene homolog 2	34	−0.94	−2.14	−1.57	−1.08	−1.29	−1.65
*Grin1*	glutamate receptor, ionotropic, N-methyl D-aspartate 1	1	−0.79	−1.87	−1.40	−0.45	−1.41	−1.87
*Gnaq*	guanine nucleotide binding protein (G protein), q polypeptide	1345	−1.04	−1.89	−1.46	−0.21	−1.19	−1.87
*Chrna7*	cholinergic receptor, nicotinic, alpha 7 (neuronal)	3	−1.38	−2.01	−1.57	−0.91	−1.36	−1.87
*Cyp2c11*	cytochrome P450, subfamily 2, polypeptide 11	5	0.54	−1.33	−0.99	0.15	−1.04	−1.86
*Gnb3*	guanine nucleotide binding protein (G protein), beta polypeptide 3	12345	−0.88	−2.24	−1.57	−0.96	−1.56	−1.81

**Table 4 cimb-47-00380-t004:** The Gene Commanding Height (CGH) scores of the most influential neurotransmission genes compared to those of the Gene Master Regulators in all six groups of profiled samples.

			Male			Female		
Gene	Description	PATH	SN	BN	BY	SN	BN	BY
*Pik3r2*	phosphoinositide-3-kinase, regulatory subunit 2	3	20	4	1	2	3	1
*Gng12*	guanine nucleotide binding protein (G protein), gamma 12	12345	8	2	8	5	1	8
*Ppp1ca*	protein phosphatase 1, catalytic subunit, alpha isozyme	4	7	9	4	2	6	4
*Ppp2r1a*	protein phosphatase 2, regulatory subunit A, alpha	4	7	3	2	3	1	2
*Calml4*	calmodulin-like 4	4	6	2	3	2	1	3
*Abat*	4-aminobutyrate aminotransferase	2	3	4	5	26	6	5
*Pld1*	phospholipase D1	1	1	1	2	17	1	2
*Slc6a7*	solute carrier family 6 (neurotransmitter transporter), member 7	0	4	1	4	11	1	4
*Gabra1*	gamma-aminobutyric acid (GABA) A receptor, alpha 1	2	1	2	4	8	2	4
*Homer1*	homer homolog 1	1	1	2	13	8	2	13
*Gphn*	gephyrin	2	3	16	0	3	2	0
*Ap2a2*	adaptor-related protein complex 2, alpha 2 subunit	0	1	16	3	2	2	3
*Creb3l1*	cAMP responsive element binding protein 3-like 1	34	1	14	4	3	1	4
*Gng10*	guanine nucleotide binding protein (G protein), gamma 10	12345	2	11	4	2	3	4
*Gabbr1*	gamma-aminobutyric acid (GABA) B receptor 1	2	1	10	5	5	2	5
*Grm8*	glutamate receptor, metabotropic 8	1	2	1	5	3	14	5
*Cpne3*	copine III	0	2	2	2	1	14	2
*Atp6v1h*	ATPase, H+ transporting, lysosomal V1 subunit H	0	4	3	6	2	12	6
*Atp6v0e1*	ATPase, H+ transporting, lysosomal, V0 subunit e1	0	3	1	1	3	12	1
*Gabarapl2*	GABA(A) receptor-associated protein like 2	2	3	8	7	1	10	7
*Mapk10*	mitogen activated protein kinase 10	4	2	6	19	1	1	19
*Homer1*	homer homolog 1	1	1	2	13	8	2	13
*Th*	tyrosine hydroxylase	4	1	1	11	2	1	11
*Gnb4*	guanine nucleotide binding protein (G protein), beta polypeptide 4	12345	3	5	10	6	1	10
*Glul*	glutamate-ammonia ligase	12	1	1	10	3	5	10
*Mapk10*	mitogen activated protein kinase 10	4	2	6	19	1	1	19
*Homer1*	homer homolog 1	1	1	2	13	8	2	13
*Th*	tyrosine hydroxylase	4	1	1	11	2	1	11
*Gnb4*	guanine nucleotide binding protein (G protein), beta polypeptide 4	12345	3	5	10	6	1	10
*Glul*	glutamate-ammonia ligase	12	1	1	10	3	5	10
*Erg28*	ergosterol biosynthesis 28 homolog		40	6	1	3	7	1
*Erap1*	endoplasmic reticulum aminopeptidase 1		2	2	2	70	2	2
*Tmem238*	transmembrane protein 238		4	162	3	3	4	3
*Taf8*	TAF8 RNA polymerase II, TATA box binding protein (TBP)-associated factor		1	1	3	5	38	3
*Tmem134*	transmembrane protein 134		2	7	78	5	2	78
*Tmem134*	transmembrane protein 134		2	7	78	5	2	78

## Data Availability

The wet protocol and the microarray raw data were deposited in the publicly accessible Gene Ex-pression Omnibus: https://www.ncbi.nlm.nih.gov/gds/?term=gse123721, https://www.ncbi.nlm.nih.gov/gds/?term=gse124613, https://www.ncbi.nlm.nih.gov/gds/?term=gse128091 (accessed on 1 April 2025).

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
