# Peer review of "Neurotransmission Sex Dichotomy in the Rat Hypothalamic Paraventricular Nucleus in Healthy and Infantile Spasm Model"

_cimb, 2025, doi:10.3390/cimb47050380_

Round 1
Reviewer 1 Report
Comments and Suggestions for Authors
The research article is well-planned. The study is novel and presents strong results supported by a robust discussion. All the figures are clearly presented, effectively supporting the results and making them easy to understand. Overall, the article does not have any scientific flaws.
Author Response
The research article is well-planned. The study is novel and presents strong results supported by a robust discussion. All the figures are clearly presented, effectively supporting the results and making them easy to understand. Overall, the article does not have any scientific flaws.
Thank you so much for the kind appreciation of our work.
Reviewer 2 Report
Comments and Suggestions for Authors
The study provides valuable insights into sex-specific transcriptomic mechanisms in IESS. It may be suitable for publication after larget revisions to enhance clarity and depth.
Q1: Keywords: Replace the colon after "glutamatergic synapse" with a semicolon for consistency:
Q2: "weighted" → should be "weighed" (Methods, Animals section: "pups were weighted and sexed").
Q3: "the brain were removed" → should be "the brains were removed"
Q4: Figure/Table References: In the main text, Figure 2 and Table 1 are referenced before their actual placement. Ensure figures/tables appear after their first mention in the text.
Q5: Please define MSN, FSN, MBN, FBN, MBY, and FBY
Q6: The equations (1)–(6b) are dense and lack sufficient textual explanation. Add a brief summary of the Genomic Fabric Paradigm (e.g., how AVE, REC, and COR collectively define transcriptomic topology).
Q7: page 7: IESS= infantile epilepsy spasm syndrome; What does mean IS in "IS female PVN"? . It should be consistently "IESS" to avoid confusion.
Q8: Methods: Please briefly explain biological meaning first (e.g., explain AVE, REC, COR simply before giving equations).
Q9: Figure 2 The caption mentions "COLOR CODE" but does not define what red/green/yellow signify in the main text. Add a legend within the figure or caption. Please also add patterns or symbols (e.g., ↑, ↓) to ensure accessibility.
Q10: In Figure 3, the opposite regulation of Atp2a2 and Cplx2 between sexes warrants deeper biological context (e.g., implications for calcium signaling or synaptic plasticity).
Q11: Tables S2a–S4 use red/green highlighting for ratios. Include footnotes explaining these color codes (e.g., "Red: higher in males; Green: higher in females").
Q12: Define specialized terms like "genomic fabrics" and "Gene Commanding Height (GCH)" in the Introduction or Methods to aid broader readership.
Q13: Results: Overreliance on statistical criteria without biological validation.
Q14: Results: No qPCR validation or independent confirmation of key findings from microarray (common practice in transcriptomics papers).
Q15: Discussion: The link between Tmem134 and its proposed role in IESS pathology is underdeveloped. Expand on why this gene’s dominance is surprising and how it relates to cytokine pathways.
Q16: Address the small sample size (n=12 per sex) as a potential limitation affecting statistical power.
Q17: Limited discussion of how prenatal betamethasone priming alone (without NMDA) affects neurotransmission pathways.
Q18: Over-reliance on transcriptomic data without protein-level validation (acknowledged but merits emphasis as a key limitation).
Q19: Authors assume transcript changes imply functional neurotransmission changes, but no direct physiological or electrophysiological validation is presented, such as functional outcomes (seizures, EEGs, behavior), and no hormonal measurements (e.g., testosterone/estrogen levels). This could be noted as a limitation.
Author Response
The study provides valuable insights into sex-specific transcriptomic mechanisms in IESS. It may be suitable for publication after large revisions to enhance clarity and depth.
We appreciate your very thorough, supporting, and constructive comments
Q1: Keywords: Replace the colon after "glutamatergic synapse" with a semicolon for consistency:
Thank you. Corrected
Q2: "weighted" → should be "weighed" (Methods, Animals section: "pups were weighted and sexed").
Thank you. Corrected
Q3: "the brain were removed" → should be "the brains were removed"
Thank you. Corrected.
Q4: Figure/Table References: In the main text, Figure 2 and Table 1 are referenced before their actual placement. Ensure figures/tables appear after their first mention in the text.
Thank you. Corrected.
Q5: Please define MSN, FSN, MBN, FBN, MBY, and FBY
Thank you. Now better explained.
Q6: The equations (1)–(6b) are dense and lack sufficient textual explanation. Add a brief summary of the Genomic Fabric Paradigm (e.g., how AVE, REC, and COR collectively define transcriptomic topology).
The equations are now better explained.
Q7: page 7: IESS= infantile epilepsy spasm syndrome; What does mean IS in "IS female PVN"? . It should be consistently "IESS" to avoid confusion.
Thank you. Corrected.
Q8: Methods: Please briefly explain biological meaning first (e.g., explain AVE, REC, COR simply before giving equations).
Thank you. Biological meaning of AVE, REC and COR are now better explained
Q9: Figure 2 The caption mentions "COLOR CODE" but does not define what red/green/yellow signify in the main text. Add a legend within the figure or caption. Please also add patterns or symbols (e.g., ↑, ↓) to ensure accessibility.
Legend better explained
Q10: In Figure 3, the opposite regulation of Atp2a2 and Cplx2 between sexes warrants deeper biological context (e.g., implications for calcium signaling or synaptic plasticity).
We added the following to the Discussion: "For instance, while in males, the ATPase Atp2a2 (a.k.a. Serca2), involved in actively pumping Ca2+ from the cytosol into the endoplasmic reticulum and a candidate gene for IESS [74] is down-regulated, it is up-regulated in females. We can speculate that this calcium homeostasis regulator [Nakajima et al., 2021] thus may be responding to different cytosolic calcium conditions after betamethasone/spasms exposure in males and females. While ATP2A2 is a monogenic cause of Darier disease (a skin disorder with neuropsychiatric abnormalities), prevalence of epilepsy in this condition is higher than in general population [Gordon-Smith et al., 2010]. Interestingly our unpublished in vitro data (not stratified for sex) indicate that hippocampal LTP is diminished after prenatal betamethasone exposure and this effect is independent of NMDA-induced spasms, increasing number of subjects and stratification by sex may fully reveal biological importance of this finding. The differences between the two sexes appear also in the regulomes of SVC (Figure 3), GLU (Figure 4) and GABA (Figure 5) pathways after induction of IS. For instance, the presynaptic (synaptic vesicle) regulatory protein Cplx2 (complexin 2), whose variants affect cognition and memory in patients with schizophrenia [81][Haas et al., 2014], is down-regulated in males (Fig. 3a) but upregulated in females (Fig. 3b). A very recent study looking at the percentage reads of new isoforms identified new CPLX2 isoforms not only in patients with schizophrenia, but also in patients with epilepsy [Heberle et al., 2024]. Our finding of altered Cplx2 expression is indirectly corroborated with a study on chronic mild stress that links Cplx2 to anxiety-susceptible experimental rat group [Liao et, 2021]."
Q11: Tables S2a–S4 use red/green highlighting for ratios. Include footnotes explaining these color codes (e.g., "Red: higher in males; Green: higher in females").
Thank you. Included.
Q12: Define specialized terms like "genomic fabrics" and "Gene Commanding Height (GCH)" in the Introduction or Methods to aid broader readership.
Thank you. Defined in the Introduction and better discussed.
Q13: Results: Overreliance on statistical criteria without biological validation.
We discussed the limitations of our investigation and the indirect validation of the sex dichotomy through behavioral and electrophysiological studies
Q14: Results: No qPCR validation or independent confirmation of key findings from microarray (common practice in transcriptomics papers).
Although a common practice, qPCR is not really a “golden standard” because, like microarrays, it is also affected by technical noise (about 12%). We have considered the technical noise in the cut-off criterion of the absolute fold-change. However, in other studies, we verified our protocol by qRT-PCR (e.g. doi: 10.1152/physiolgenomics.00217.2004).
Q15: Discussion: The link between Tmem134 and its proposed role in IESS pathology is underdeveloped. Expand on why this gene’s dominance is surprising and how it relates to cytokine pathways.
We added the following: "However, present study pinpoints Tmem134, a protein located in the perinuclear region of the cytoplasm, involved in RNA splicing [92] and associated with obesity [93], as the best target for the gene therapy of IESS in rats of both sexes based on symmetrical expression change in both sexes. Additionally, our model suggests significant involvement of hypothalamic nuclei such as the arcuate nucleus [7], also participating in the control of food intake. Tmem134 is also involved in the regulation of NF-kappaB pathway [Tian et al., 2017]. Activity in this pathway is increased after seizures and downregulation of this pathway attenuates seizures [Cai and Lin, 2022]. Hence affecting this pathway through Tmem134 may provide a feasible way for control of spasms."
Q16: Address the small sample size (n=12 per sex) as a potential limitation affecting statistical power.
Sample size of 12 is too low to provide reliable stratification by sex, especially if biological variables such as spasms or body weight are considered. However, as mentioned above, our unpublished and published electrophysiology data [Benson et al, 2020] as well as a behavioral study [Velisek, 2006] indicate electrophysiological as well as behavioral effects of prenatal betamethasone exposure, though larger sample sizes would be required for proper stratification by sex. Finally, the effects of repetitive spasms during development (P12, P13, P15) on behaviors and delayed seizure susceptibility are still being investigated and therefore are missing as comparators. Most genomists use 3 biological replicas per condition, we used 4 (x 3 conditions = 12 per sex) that considerably increased the statistical relevance. More than 4 replicas would bring no advantage because of the technical noise.
Q17: Limited discussion of how prenatal betamethasone priming alone (without NMDA) affects neurotransmission pathways.
We extended the discussion on the effects of the betamethasone priming alone (without NMDA) on the neurotransmission pathways
Q18: Over-reliance on transcriptomic data without protein-level validation (acknowledged but merits emphasis as a key limitation).
We are aware that gene transcription and transcript translation into protein exhibit a complex correlation, something like acceleration and speed of a vehicle: the gene is transcribed when the cell needs to increase or replenish the protein content. Since in addition to translation, protein abundance is affected by several post-translational modifications, and the genes’ expressions correlations do not exactly overlap the protein-protein interactions our results are relevant only at the transcriptomic level and integration with epigenetic and proteomic data are needed to strength the conclusion.
Q19: Authors assume transcript changes imply functional neurotransmission changes, but no direct physiological or electrophysiological validation is presented, such as functional outcomes (seizures, EEGs, behavior), and no hormonal measurements (e.g., testosterone/estrogen levels). This could be noted as a limitation.
Actually, on this rat model, we have performed and published numerous other than genomic experiments, some of them cited in this manuscript: electroencephalography [7, 8, 23], behavior [23, 25], immunohistochemistry [42], neuronal counting [42], slice blotting [42]
Reviewer 3 Report
Comments and Suggestions for Authors
The paper describes sex dichotomy in the rat hypothalamic paraventricular nucleus in health and infantile spasm model on the basis of 2018 and 2019 deposited expression array data. I have the following critical comments:
- The authors must provide much greater details about the performed expression analysis. So it is not which expression arrays have been used in this study. The description in §2.2 is completely insufficient.
- To convince the critical reader of the manuscript the authors should consider to repeat their analysis with RNAseq which is the present state-of-the-art method for expression profiling.
- The numbers presented in the tables remain complete unclear. The provided legends are very short and not informative.
- The authors should provide an information about the variability of their data (SD or SEM). That cannot be seen from the provided table.
- The authors should express the novelity in respect to the study PMID: 29636502
Author Response
- The authors must provide much greater details about the performed expression analysis. So it is not which expression arrays have been used in this study. The description in §2.2 is completely insufficient.
Thank you. The text now presents the features of the used Agilent-028282 Whole Rat Genome Microarray 4x44K v3 whose full description is included in the cited GEO databases.
- To convince the critical reader of the manuscript the authors should consider to repeat their analysis with RNAseq which is the present state-of-the-art method for expression profiling
At the time of the experiments (2018), we found by profiling technical replicates that, in addition to higher costs, Illumina MiSeq and NextSeq 550 had also higher technical noise (~30%) that the Agilent microarrays with our optimized wet protocol (~17%). Therefore, we fond little incentives to validate a noisy experiment with another even noisier with disputable scientific benefits.
- The numbers presented in the tables remain complete unclear. The provided legends are very short and not informative
Thank you. The legends were extended to be more informative
- The authors should provide an information about the variability of their data (SD or SEM). That cannot be seen from the provided table.The variability of the expression level was included in the REV, the chi-square mid-interval estimate of the coefficient of variation, all other measures incorporating the variability by definition.
- The authors should express the novelity in respect to the study PMID: 29636502
In PMID: 29636502 we profiled the hypothalamic arcuate nuclei of male and female rat pups subjected to the same conditions, while here we analyzed expression data from the paraventricular nuclei. As expected, in addition to performing different neurologic functions, distinct nuclei had also distinct gene expression profiles and presented substantial sex differences in neurotransmission transcriptomes.
Round 2
Reviewer 2 Report
Comments and Suggestions for Authors
accepted
Author Response
Thank you for accepting our work
Reviewer 3 Report
Comments and Suggestions for Authors
The authors have addressed some of my comments.
1. However they failed to convincingly explain in the manuscript which advance this study makes in addition to their published study PMID: 29636502 and why data from 2018 were used.
2. The description of the method is still not sufficient and no experimental proof is given why these data should be better than state-of-the-art semiquatitative RNAseq data.
3. The tables still do not contain any information about the variability of data.
Author Response
- However they failed to convincingly explain in the manuscript which advance this study makes in addition to their published study PMID: 29636502 and why data from 2018 were used.We still believe that profiling a different hypothalamic nucleus, that responsible for a distinct function in the brain was not a waste of time
- The description of the method is still not sufficient and no experimental proof is given why these data should be better than state-of-the-art semiquatitative RNAseq data.The experimental protocol is fully described in the cited GEO deposits. As the Director of the Personalized Genomics Laboratory, DAI had fully access to both Agilent microarrays and Illumina RNA-sequencing platforms. However, by profiling technical replicas (that nobody does because of the high costs) we found that at the time the sequencing was not only more expensive but also noisier.
- The tables still do not contain any information about the variability of data As mentioned in the main text, the variability of the expression data is incorporated in the independent (with respect to the average expression level) measure relative expression control.